# Anthropomorphism of Nature, Environmental Guilt, and Pro-Environmental Behavior

**Kim-Pong Tam** 

Division of Social Science, The Hong Kong University of Science and Technology, Hong Kong, China; kevintam@ust.hk

**Abstract:** Feeling guilty about the occurrence of environmental problems is not uncommon; however, not everyone experiences it. Why are there such individual differences? Considering that guilt is a predominantly interpersonal phenomenon, as emotion research has demonstrated, how is it possible that some individuals feel guilty for the degradation of the non-human environment, and some others do not? The present investigation tests an integrated solution to these two questions based on the concept of anthropomorphism. In three studies, with an individual difference approach, it was observed that anthropomorphism of nature predicted the experience of environmental guilt, and this feeling in turn was associated with engagement in pro-environmental behavior. That is, it appears that individuals who view nature in anthropomorphic terms are more likely to feel guilty for environmental degradation, and they take more steps toward environmental action. This observation not only improves existing understanding of environmental guilt, but also adds evidence to the theoretical possibility of describing and understanding the human–nature relationship with reference to psychological knowledge regarding interpersonal relationships.

**Keywords:** guilt; emotion; pro-environmental behavior; anthropomorphism; human-nature relationship

---

## 1. Introduction

Feeling guilty for what is going wrong in the environment is not an uncommon experience. In *Greendex*, a study that explored consumer behavior in 18 countries [1], about one third of the 18,000 respondents agreed or strongly agreed with the statement "I feel guilty about the impact I have on the environment". Even devoted environmentalists who actively practice green living report experiencing guilt on occasion [2]. Advice on how to overcome environmental guilt is abundant on the Internet and in the book market [3]. Social innovations that allow consumers to atone for their guilt are emerging [4]. Annina Rüst, a Swiss-born artist-inventor, even designed a leg band that would drive stainless-steel thorns into the user's leg when he/she is using too much electricity; Rüst referred to this self-mortification device as a "therapy for environmental guilt" [5].

Notably, critiques regarding the experience of environmental guilt are plentiful. Some people question whether feeling guilt for environmental problems is meaningful; some even put forward that guilt is being used manipulatively by environmentalists and could be counterproductive [6]. Interestingly, in *Greendex*, another third of the respondents were opposed to the feeling of environmental guilt, with the remaining third taking a neutral stance [1]. What is the reason behind these individual differences?

There is another unresolved issue regarding the experience of environmental guilt. Emotion research has suggested that guilt is predominantly experienced when an individual perceives that his/her own action has caused some harm to another person [7]. As Baumeister, Stillwell, and Heatherton [8] concluded in their seminal review, guilt is essentially an interpersonal phenomenon. This notion creates an inconvenience for the understanding of environmental guilt: If guilt is indeed an interpersonal



emotion, how is it possible for some individuals to experience guilt for the degradation of the non-human environment?

In this investigation, I address these two issues with reference to the concept of anthropomorphism [9,10]. I argue that anthropomorphism of nature allows some individuals to attribute moral patiency, an indispensable component in the appraisal of harm, to the environment, and thereby experience guilt in the face of its degradation. In this article, I report a collection of supportive evidence for this argument based on three studies.

## 1.1. Environmental Guilt

The feeling of guilt typically motivates apologies, attempts to reduce the harm, and actions to make up for one's misdeeds and compensate for the victim's loss [11,12]. Considering these behavioral consequences, many researchers and practitioners are intrigued by the potential utility of guilt appeals in environmental communication. That is, if we can induce a sense of guilt in the public for the occurrence of environmental problems, chances are, people will desire to make amends and thereby adopt a more environment-friendly lifestyle [13].

Studies have shown that environmental guilt is indeed associated with pro-environmental behavior [14]. For instance, an early study [15] revealed a small to moderate correlation ($r = 0.23$, or 0.33 after correction for measurement error) between environmental guilt and general ecological behavior. The effect of this size was corroborated in a meta-analytic review that examined the influences of various social cognitive factors [16]; this review revealed an overall moderate effect of environmental guilt (95% CI: 0.21, 0.38, based on five studies). Recent studies have also expanded on these earlier findings by showing that people may experience guilt for the ingroup's (rather than personal) contribution to environmental problems, and this form of guilt, or collective guilt, also motivates pro-environmental actions. For example, a study [17] showed that guilt for the ingroup's greenhouse gas emissions was associated with willingness to conserve energy and pay green taxes. Another study [18] showed that Americans who reported a stronger feeling of guilt for their country's contribution to environmental problems and lack of actions to protect the environment reported stronger willingness to engage in eco-friendly behavior.

## 1.2. The Interpersonal Essence of Guilt

Psychological research on emotions has shown that guilt is prototypically experienced when an individual perceives that his/her own action has caused some harm to another person [7]. As a study revealed [19], research participants predominantly reported feeling guilty about events that involved interpersonal harm. In another study [12], when asked to recall a situation in which they had experienced guilt, 87.5% of participants reported a situation that involved harm done to another person. This association between guilt and interpersonal harm seems to be cross-culturally robust [20].

As Baumeister et al. [8] concluded in their seminal review, guilt is essentially an interpersonal phenomenon. Guilt typically involves concern over exclusion by other people and very often motivates behavior that can repair or enhance social relationships (e.g., apology, reparations, compensations). This interpersonal account is in line with how evolutionary theorists understand guilt [8]: Human beings have developed the experience of guilt because it helps prevent them from damaging relationships with others, which are essential for survival.

The interpersonal essence of guilt creates an inconvenience for the understanding of environmental guilt: If guilt is indeed an interpersonal emotion in essence, how is it possible that some individuals feel guilty for the degradation of the non-human environment?

One way to address this question is to argue that people actually feel guilty not for the harm done to the environment but for the harm caused to other people affected by environmental problems (e.g., future generations, populations in the Third World who are most affected by climate change). In this view, the experience of environmental guilt is triggered by the suffering of some humans, which is still in line with the interpersonal essence of guilt. Another possible solution is to assume that guilt

is not exclusively interpersonal. Even though guilt predominantly arises in interpersonal contexts, it may still arise in situations that do not involve any interpersonal harm (e.g., failure to meet personal standards) [12]. In the present research, with reference to the recent theoretical developments in the study of morality and research of anthropomorphism, I propose a third solution, a view that has been overlooked in previous studies.

### 1.3. Anthropomorphism of Nature and Environmental Guilt

The theory of dyadic morality [21,22] states that all moral judgments and subsequent emotional reactions are rooted in the cognitive template of harm. This template involves two connected minds, or a moral dyad, that includes an intentional agent, who causes the harm, and a sentient patient, who receives the harm. Because different people may have different perceptions regarding moral agency and moral patiency in the same event, they experience different moral judgments and hence different emotional outcomes. For example, because there are individual and cultural differences in the perception of the level of moral patiency in fetuses and animals [23], there are different moral judgments and emotional reactions with regard to abortion and animal use [22]. For the same logic, murder is usually considered to be immoral and typically arouses anger in observers; however, in war combats, killing is typically seen as morally acceptable and justifiable because enemies are very often perceived as less human or even non-human (i.e., dehumanized), and thereby denied moral patiency [24].

Viewed from the theory of dyadic morality, it is not impossible for a person to feel guilty for the degradation of the non-human environment. That is, when the person perceives moral patiency in the environment, the cognitive template of harm will apply, and environmental guilt will likely follow. Attributing moral patiency to the environment (that is, to consider it as sentient and able to suffer) obviously deviates from the objective reality. However, just as people may dehumanize other humans [24], which is ubiquitous, subjectively real, and psychologically impactful, they may humanize or anthropomorphize non-humans [9,25]. Studies have shown that it is not uncommon for people to attribute human qualities (e.g., mental capacities) to such non-human entities as cars [26] and slot machines [27]. Recent studies have also shown that people are more likely to comply with a request from a social cause when it is anthropomorphized [28]. Most relevant to the present investigation is the finding that some individuals tend to spontaneously consider the natural world as humanlike [10,25]. Moreover, recent studies have demonstrated that individuals who anthropomorphize nature tend to experience supposedly interpersonal processes, such as social connectedness [10] and empathy [29,30], toward the environment. Overall, with anthropomorphism, people can relate to the environment in a way that is similar to how they relate to other humans. In other words, in their relationship with the natural world, it is not impossible for people to have experiences that are characteristics of interpersonal relationships [10], which include the experience of guilt.

Based on the reasoning above, I argue that the cognitive tendency of anthropomorphism of nature, which activates the perception of moral patiency in the non-human environment, allows individuals to experience guilt in the face of environmental degradation. This argument can also address the other question regarding environmental guilt raised earlier: What explains the individual differences of the experience of environmental guilt? Based on the reasoning above, it is expected that individual differences in the cognitive tendency of anthropomorphism of nature are associated with individual differences in environmental guilt.

### 1.4. The Present Research

In sum, I hypothesize a psychological pathway from anthropomorphism of nature, through environmental guilt, to pro-environmental behavior. That is, due to the individual differences in anthropomorphism of nature, some individuals are more likely than others to experience environmental guilt, and as a result they are more motivated to take pro-environmental behavior.

I tested this hypothesis with three studies. Acknowledging the increasing concern of replicability of research findings in psychology [31], when conducting these three studies, I attempted to diversify the

study designs, measures, and samples in order to establish the robustness of my findings. To address the potential threats of the "third variable" problem [32] and the item context effects [33], across the studies, I controlled for the influence of several factors (e.g., gender, general pro-environmental worldview) that are known to be associated with anthropomorphism on the one hand and pro-environmental behavior on the other hand, and made use of multiple measures of the same constructs. To establish the generalizability of the findings, I used participants from different age groups (students and working adults) and different cultural backgrounds (Hong Kong Chinese and British). I will discuss these methodological issues in greater detail at the beginning of each study.

Before moving to the report of the studies, I would like to keep the objective of the present investigation in perspective. The goal of the present research is to empirically spell out the relationship between anthropomorphism of nature (which is a pervasive cognitive tendency) and the experience of environmental guilt (which is not uncommon). Although apparently a pro-environmental effect of anthropomorphism is hypothesized, findings from the present research should *not* be taken as an advocate for the use of anthropomorphism to facilitate environmental conservation. It is noteworthy that in the scientific discourse, there have been opposing views to the use of anthropomorphism. One concern is that anthropomorphism is infantile and just an immature form of reasoning, and it therefore will not be sustainable among adults (although this view to some extent has been rebutted by findings showing that a substantial proportion of the adult participants did exhibit dispositional anthropomorphism [34]). Another concern is that cultivating anthropomorphism may obfuscate people's objective, accurate understanding and representations of the world [35,36]. In addition, there have been suggestions that anthropomorphizing animals or nature may lead to sensationalized and romanticized depictions of wildlife and natural environments that may actually bear ill impacts on conservation (e.g., overly simplifying conservation issues) [37]. For example, the "Bambi effect" may have cultivated the ethic of non-intervention when it comes to the management of the wild deer population, and this attitude may bring inadvertent and undesired impacts on other species and forests [38]. Readers should take note of these views before drawing practical implications from the present research. I will return to this issue later.

## 2. Study 1

Study 1 was embedded in a broader investigation regarding psychological antecedents of participation in Earth Hour. Earth Hour is a globally celebrated environmental campaign initiated by the World Wide Fund for Nature; its vision is to inspire people around the world to "switch from passive bystanders to active participants in global efforts to fight climate change" [39]. Every March, people share their commitment to the environment through the symbolic act of turning lights off for an hour. Thus, participation in the event is assumed to be a valid indicator of pro-environmental behavior. In this study, with a sample of working adults in Hong Kong, I tested whether anthropomorphism of nature was associated with environmental guilt, and whether guilt, in turn, was associated with self-report participation in the event in the past and intention to participate in it in future.

Age and gender are known to be associated with anthropomorphism of nature [29,40] and pro-environmental behavior [41]. To address the potential threat of their confounding effect, they were controlled for in the main analysis.

### 2.1. Participants

The sample comprised 176 adult staff members (73 males and 103 females) from a university in Hong Kong. To determine the sample size required for 0.80 statistical power, I needed to estimate the size of the associations of interest. First, based on previous studies [16], I assumed the size of the association between environmental guilt and pro-environmental behavior to be small to moderate. The size of the association between anthropomorphism of nature and environmental guilt has never been estimated in previous studies; however, based on past studies regarding anthropomorphism of nature [29,30], I assumed the size of this association to be small to moderate too. With these two

effect sizes, to test the hypothesized mediation effect (i.e., guilt mediating the association between anthropomorphism and behavior) with the bias-corrected bootstrap test, a sample size of 148 is required for 0.80 power [42]. The achieved sample size of 176 was sufficient to meet this requirement.

Participants' age ranged from 22 to 60 years (mean = 32.86 and S.D. = 8.37 years; 11 unreported). They participated in a broader study titled "Survey Study on Earth Hour." They were compensated HKD40 (approximately USD5) for their participation.

## 2.2. Measures

The measures relevant for the present study included the following. In the Anthropomorphism of Nature Scale (ANS) [29,30,40], participants answered five questions: "To what extent does nature have a mind of its own/intentions/free will/consciousness/emotional experience?" on an 11-point scale (0 = not at all to 10 = very much; $\alpha = 0.97$). Environmental guilt was measured by two items written with reference to past studies [43]. The items read: "I am regretful about humans' lack of performance on environmental protection", and "I feel guilty for how little humans have done to improve the quality of the environment". Participants responded on a 7-point scale (1 = strongly disagree to 7 = strongly agree; $\alpha = 0.88$). As for participation in Earth Hour, two measures were used. The first one referred to participants' intention to turn lights off during Earth Hour. There were four items (e.g., "I intend to turn off non-essential lights during Earth Hour this year"). Participants responded on a 7-point scale (1 = strongly disagree to 7 = strongly agree; $\alpha = 0.95$). The second measure referred to various forms of participation in the event in the past (e.g., "made a pledge on the official Earth Hour website", "discussed Earth Hour with your family or friends"). Participants indicated whether they had ever done each of these actions by indicating Yes (1) or No (0) ($\alpha = 0.67$).

A mean score was computed for each measure (except past participation, for which a sum score was computed). The presentation order of all measures was randomized across participants. (Note: All measures and data supporting the analyses in this study and the subsequent studies are included in the Supplemental Materials.).

## 2.3. Results and Discussion

Table 1 shows the correlations among the variables. As hypothesized, anthropomorphism was positively associated with guilt. Furthermore, anthropomorphism was positively associated with both past participation in Earth Hour and intention to participate in it in future. Guilt was positively associated with both Earth Hour participation measures. All correlations were significant.

**Table 1.** Descriptive statistics and inter-correlations in Study 1.

|  |  | Mean (S.D.) | 1 | 2 | 3 |
|---|---|---|---|---|---|
| 1. | ANS | 4.14 (1.67) |  |  |  |
| 2. | Guilt | 5.11 (1.08) | 0.25 ** |  |  |
| 3. | Intention to participate in future | 5.44 (1.31) | 0.23 ** | 0.40 *** |  |
| 4. | Participation in the past | 2.54 (1.65) | 0.17 * | 0.32 *** | 0.33 *** |

*Notes.* *** $p < 0.001$; ** $p < 0.01$; * $p < 0.05$. ANS = Anthropomorphism of Nature Scale.

I then tested the hypothesized pathway from anthropomorphism, through guilt, to Earth Hour participation. Both age and gender were controlled for. Table 2 shows the results. Anthropomorphism significantly predicted guilt (Path a). It also significantly predicted both measures of Earth Hour participation (Path c). When anthropomorphism and guilt were both included in the model predicting Earth Hour participation (Path c' and Path b, respectively), only guilt was a significant predictor.

With the bias-corrected bootstrap test (5000 resamples) in the SPSS PROCESS macro [44], I observed that the indirect effect of anthropomorphism through guilt (a X b) was significant (i.e., the 95% CI did not include zero).

**Table 2.** Results of regression analyses and test of indirect effect in Study 1.

| | | *Outcome in the Model* | |
| | | **Intention to Participate in Future** | **Participation in the Past** |
| --- | --- | --- | --- |
| | *Unstandardized regression coefficients (standard errors)* | | |
| Path a | ANS → guilt | 0.16 *** (0.05) | 0.16 *** (0.05) |
| Path b | guilt → PEB | 0.42 *** (0.09) | 0.46 *** (0.12) |
| Path c | ANS → PEB (total effect) | 0.15 * (0.06) | 0.19 * (0.08) |
| Path c′ | ANS → PEB (direct effect) | 0.08 (0.06) | 0.11 (0.08) |
| | *Size of indirect effect (bootstrap test 95% CI)* | | |
| a X b | ANS → guilt → PEB | 0.069 (0.02, 0.16) | 0.076 (0.02, 0.17) |

Notes. *** $p < 0.001$; ** $p < 0.01$; * $p < 0.05$. Path a = anthropomorphism to guilt. Path b = guilt to PEB (when anthropomorphism was also in the model). Path c = anthropomorphism to PEB (i.e., total effect). Path c′ = anthropomorphism to PEB (when guilt was also in the model; i.e., direct effect). Age and gender were controlled for. ANS = Anthropomorphism of Nature Scale. PEB = pro-environmental behavior.

In summary, the findings provide preliminary support to the hypothesis. Participants who anthropomorphized nature to a larger extent experienced more environmental guilt; this stronger feeling of guilt, in turn, was associated with more intense participation in Earth Hour in the past and a stronger intention to participate in it in future.

## 3. Study 2

In Study 2, I attempted to replicate Study 1 with some methodological improvements.

First, the emotion measure in Study 1 did not specify any environmental problems; I relied on participants' knowledge about these problems. It is likely that there were substantial individual differences in terms of environmental knowledge, and that different participants thought about different environmental problems. Some participants might have reported higher levels of guilt just because they were considering more global and serious problems. To remove these extraneous influences, in Study 2, I standardized the environmental problems by presenting to participants the same set of photos depicting such problems.

Second, in Study 1, no other emotions were considered. In Study 2, I additionally included anger, anxiety, and shame. It is possible that people feel angry when they focus on other people's contribution to environmental problems; however, such emotion was not expected to motivate any reparative, pro-environmental actions [45]. People may feel anxious in the face of environmental problems as they may fall victims to such problems in future, and environmental anxiety also motivates pro-environmental behavior as a means of coping [14]. When considering one's own contribution to environmental problems, people may feel ashamed because they perceive that such misdeeds reflect something globally wrong about themselves (e.g., flawed character, corrupt values) [18]. Notably, while guilt typically triggers reparative behavior, shame tends to activate withdrawal, emotion-focused behavior (e.g., denial, avoidance) [46]. Accordingly, it was expected that shame would not be associated with pro-environmental behavior. On the whole, it was expected that the hypothesized pathway through guilt would still hold even when the other three negative emotions were controlled for. Such findings would support the validity of the hypothesized pathway.

Third, it is widely recognized that pro-environmental behavior is not a singular concept [47]. Some behavior occurs in people's private sphere of life (e.g., energy saving); this behavior typically has direct, though small, environmental impact. Some behavior tends to be organized and occur in the public sphere (e.g., environmental protests); its impact on the environment depends on other people's participation and is typically indirect (through influencing others and governments). Notably, most studies on environmental guilt have focused on one type of pro-environmental behavior only [43,46].

Earth Hour participation in Study 1 apparently fell into the latter type. In Study 2, I measured both types of pro-environmental behavior.

Fourth, to further remove the potential threat of the "third variable" problem, in addition to age and gender, participants' general pro-environmental worldview was controlled for. Such a worldview was measured with the New Ecological Paradigm (NEP) scale, a widely used measure of environmental attitude and environmental concern [48].

Fifth, it is important to establish that it is anthropomorphism of nature, not anthropomorphism of anything else, that triggers the observed effects. Thus, in addition to anthropomorphism of nature, I measured other forms of anthropomorphism and compared their effects.

Last, I adopted the prospective design. Anthropomorphism was measured at Time 1, while guilt and pro-environmental behavior were measured at Time 2 (a week later). This design was less susceptible to the biases introduced by the item context effects (e.g., predictor and outcome variables measured at the same point in time) [33], which could inflate relationships artificially. It was also consistent with the hypothesized temporal sequence of the variables.

### 3.1. Participants

The sample comprised 168 undergraduate students (76 males and 92 females) studying an introductory psychology course at a university in Hong Kong. They participated on a voluntary basis for course credits. Their age ranged from 18 to 25 years (mean = 20.73 and S.D. = 1.20 years). All participants were of Chinese ethnicity. Based on what I reported at the beginning of Study 1, this sample size was sufficient for achieving 0.80 power for the assumed effect sizes.

### 3.2. Measures

At Time 1, participants completed two measures of anthropomorphism. The first measure was the ANS ($\alpha$ = 0.96). The second one was the Individual Differences in Anthropomorphism Questionnaire (IDAQ) [34]. For each of the 15 items, participants indicated to what extent a non-human entity possessed human characteristics (e.g., free will, consciousness). Five entities were technological devices (e.g., TV set), five non-human animals (e.g., fish), and five natural entities (e.g., mountain). All responses were made on an 11-point scale (0 = not at all to 10 = very much). A factor analysis revealed a neat three-factor structure. These sub-measures were reliable ($\alpha$s = 0.87 for IDAQ-devices, 0.86 for IDAQ-animals, and 0.92 for IDAQ-nature). Participants also completed the NEP scale ($\alpha$ = 0.69) [48], which served as a control variable.

One week later (Time 2), participants returned and completed another set of measures. Participants were first presented a slideshow of 10 photos depicting various environmental problems (e.g., rivers being polluted by chemicals and toxic waste from factories, forests being cleared and degraded into wasteland). The photos were shown automatically, one by one, each for eight seconds. After viewing the photos, participants reported their feelings with regard to a list of nine emotional labels (guilt: guilt, regretful, and remorseful; anger: angry and annoyed; anxiety: anxious and worried; shame: ashamed and disgraceful). They indicated the extent to which each label described their feelings while thinking about the damage humans have caused to the environment on an 11-point scale (0 = not at all to 10 = extremely). The scale reliability of the four emotions was satisfactory ($\alpha$s = 0.85, 0.62, 0.75, and 0.76, respectively). It is noteworthy that in an exploratory factor analysis, I failed to find the expected four-factor solution for these items; instead, all items fell into one dominant factor (Eigenvalue = 5.48, explained variance = 60.86%), which seemingly represented negative feelings in general. The items were therefore grouped and scored based on their intended theoretical meanings, not the observed factor structure. This limitation would be addressed in Study 3.

Participants then completed three measures of pro-environmental behavior intention. The first measure referred to private-sphere behavior. Participants reported their intention to perform 10 actions (e.g., looking for ways to reuse things); these actions were adopted from past studies [29]. The second and third measures were the two subscales in the Environmental Action Scale [49]. All items in these

two subscales referred to actions that were collective in nature. The participatory actions subscale (10 items) focused on actions in the participatory role (e.g., taking part in a protest/rally about an environmental issue), while the leadership actions subscale (eight items) focused on actions in the leadership role (e.g., organizing an environmental protest/rally). Participants indicated their intention to perform each of these actions on an 11-point scale (0 = I certainly will not do it; to 10 = I certainly will do it). These measures were also reliable ($\alpha$s = 0.85, 0.91, and 0.92, respectively).

A mean score was computed for each measure.

### 3.3. Results and Discussion

Table 3 presents the correlations among the variables. As hypothesized, anthropomorphism of nature (ANS or IDAQ-nature) was significantly associated with guilt but not the other emotions (except shame). Anthropomorphism of nature was significantly associated with all three measures of pro-environmental behavior intention. These patterns were mostly not observed for IDAQ-devices or IDAQ-animals, suggesting that the associations with environmental guilt and pro-environmental behavior were specific to anthropomorphism of the non-human and inanimate aspects of the natural world. Guilt was significantly associated with pro-environmental behavior; the same was true for the other emotions, though apparently to a weaker extent.

I then tested the hypothesized pathway from anthropomorphism, through guilt, to pro-environmental behavior. IDAQ-devices and IDAQ-animals were no longer considered. Age, gender and NEP were controlled for. Table 4 shows the results. Anthropomorphism (ANS or IDAQ-nature) significantly predicted guilt (Path a) and marginally significantly predicted shame. It also significantly predicted pro-environmental behavior intention (Path c). When anthropomorphism, guilt and the other emotions were all included in the model (Path c' and Path b, respectively), anthropomorphism was consistently a significant predictor (except for IDAQ-nature and private-sphere behavior), and the predictive power of guilt was either significant or marginally significant. Using the bootstrap test (5,000 resamples), I observed that the hypothesized indirect effect via guilt (a X b) was significant (i.e., the 95% CI did not include zero). There was no significant indirect effect through the other emotions.

In summary, the hypothesized pathway from anthropomorphism, through guilt, to pro-environmental behavior was observed. Notably, this pathway was not observed for the other forms of anthropomorphism. Furthermore, it still held even when the other negative emotions associated with environmental problems were controlled for.

As noted earlier, the four emotions measured were not empirically separable from each other in the factor analysis. As a result, there was not an uncontaminated measure of guilt as opposed to the other emotions. This might explain why the association between guilt and pro-environmental behavior was not robust. This issue would be addressed in Study 3.



**Table 3.** Descriptive statistics and inter-correlations in Study 2.

| | Mean (S.D.) | 1 | 2 | 3 | 4 | 5 | 6 | 7 | 8 | 9 | 10 |
|---|---|---|---|---|---|---|---|---|---|---|---|
| 1. ANS | 4.41 (2.79) | | | | | | | | | | |
| 2. IDAQ-devices | 1.89 (2.00) | 0.42 *** | | | | | | | | | |
| 3. IDAQ-animals | 6.85 (1.82) | 0.35 *** | 0.14 | | | | | | | | |
| 4. IDAQ-nature | 3.86 (2.51) | 0.78 *** | 0.57 *** | 0.42 *** | | | | | | | |
| 5. Guilt | 6.26 (1.83) | 0.23 ** | −0.07 | 0.16 * | 0.20 ** | | | | | | |
| 6. Anger | 6.40 (1.86) | 0.13 | −0.08 | 0.15 | 0.11 | 0.61 *** | | | | | |
| 7. Anxiety | 7.10 (1.74) | 0.08 | −0.10 | 0.11 | 0.07 | 0.68 *** | 0.67 *** | | | | |
| 8. Shame | 6.72 (1.76) | 0.17 * | −0.04 | 0.10 | 0.14 | 0.80 *** | 0.69 *** | 0.73 *** | | | |
| 9. Private-sphere behavior | 7.19 (1.46) | 0.24 ** | −0.20 | 0.22 ** | 0.17 * | 0.46 *** | 0.43 *** | 0.45 *** | 0.41 *** | | |
| 10. Participatory actions | 4.46 (2.02) | 0.27 *** | −0.06 | 0.12 | 0.23 ** | 0.45 *** | 0.37 *** | 0.35 *** | 0.38 *** | 0.65 *** | |
| 11. Leadership actions | 2.22 (1.75) | 0.35 *** | 0.18 * | 0.02 | 0.29 *** | 0.35 *** | 0.27 *** | 0.24 ** | 0.29 *** | 0.44 *** | 0.78 *** |

*Notes.* *** $p < 0.001$; ** $p < 0.01$; * $p < 0.05$. ANS = Anthropomorphism of Nature Scale. IDAQ = Individual Differences in Anthropomorphism Questionnaire.

**Table 4.** Results of regression analyses and tests of indirect effects in Study 2.

| | | Private-Sphere Behavior | Outcome in the Model<br>Participatory Actions | Leadership Actions |
|---|---|---|---|---|
| *Unstandardized regression coefficients (standard errors)* | | | | |
| Path a | ANS → guilt | 0.13 ** (0.05) | 0.13 ** (0.05) | 0.13 ** (0.05) |
| | ANS → anger | 0.07 (0.05) | 0.07 (0.05) | 0.07 (0.05) |
| | ANS → anxiety | 0.03 (0.05) | 0.03 (0.05) | 0.03 (0.05) |
| | ANS → shame | 0.08 $^{p\,=\,0.081}$ (0.05) | 0.08 $^{p\,=\,0.081}$ (0.05) | 0.08 $^{p\,=\,0.081}$ (0.05) |
| Path b | guilt → PEB | 0.16 $^{p\,=\,0.079}$ (0.09) | 0.34 * (0.13) | 0.22 $^{p\,=\,0.063}$ (0.12) |
| | anger → PEB | 0.11 (0.07) | 0.16 (0.11) | 0.09 (0.10) |
| | anxiety → PEB | 0.18 * (0.09) | 0.05 (0.13) | 0.02 (0.11) |
| | shame → PEB | −0.08 (0.10) | −0.07 (0.15) | −0.02 (0.14) |
| Path c | ANS → PEB (total effect) | 0.10 ** (0.04) | 0.18 ** (0.05) | 0.22 *** (0.05) |
| Path c′ | ANS → PEB (direct effect) | 0.07 * (0.03) | 0.13 * (0.05) | 0.19 *** (0.05) |
| *Size of indirect effects (bootstrap test 95% CI)* | | | | |
| a X b | ANS → guilt → PEB | 0.02 (0.002, 0.06) | 0.04 (0.01, 0.11) | 0.03 (0.001, 0.08) |
| | ANS → anger → PEB | 0.008 (−0.003, 0.04) | 0.01 (−0.004, 0.05) | 0.006 (−0.004, 0.03) |
| | ANS → anxiety → PEB | 0.005 (−0.01, 0.03) | 0.001 (−0.01, 03) | 0.0006 (−0.01, 0.02) |
| | ANS → shame → PEB | −0.007 (−0.05, 0.01) | −0.006 (−0.06, 0.02) | −0.002 (−0.04, 0.02) |
| *Unstandardized regression coefficients (standard errors)* | | | | |
| Path a | IDAQ-nature → guilt | 0.14 ** (0.05) | 0.14** (0.05) | 0.14 ** (0.05) |
| | IDAQ-nature → anger | 0.08 (0.06) | 0.08 (0.06) | 0.08 (0.06) |
| | IDAQ-nature → anxiety | 0.05 (0.05) | 0.05 (0.05) | 0.05 (0.05) |
| | IDAQ-nature → shame | 0.09 $^{p\,=\,0.077}$ (0.05) | 0.09 $^{p\,=\,0.077}$ (0.05) | 0.09 $^{p\,=\,0.077}$ (0.05) |
| Path b | guilt → PEB | 0.17 $^{p\,=\,0.061}$ (0.09) | 0.35 ** (0.13) | 0.25 * (0.12) |
| | anger → PEB | 0.12 (0.08) | 0.16 (0.11) | 0.10 (0.10) |
| | anxiety → PEB | 0.16 $^{p\,=\,0.059}$ (0.08) | 0.04 (0.13) | −0.004 (0.12) |
| | shame → PEB | −0.08 (0.10) | −0.07 (0.15) | −0.02 (0.14) |
| Path c | IDAQ-nature → PEB (total effect) | 0.09 * (0.04) | 0.19 ** (0.06) | 0.20 *** (0.05) |
| Path c′ | IDAQ-nature → PEB (direct effect) | 0.06 (0.04) | 0.13 * (0.06) | 0.16 ** (0.05) |
| *Size of indirect effects (bootstrap test 95% CI)* | | | | |
| a X b | IDAQ-nature → guilt → PEB | 0.02 (0.002, 0.07) | 0.05 (0.01, 0.12) | 0.03 (0.003, 0.10) |
| | IDAQ-nature → anger → PEB | 0.009 (−0.003, 0.04) | 0.01 (−0.01, 0.06) | 0.008 (−0.004, 0.04) |
| | IDAQ-nature → anxiety → PEB | 0.008 (−0.01, 0.04) | 0.002 (−0.01, 0.03) | −0.0002 (−0.02, 0.01) |
| | IDAQ-nature → shame → PEB | −0.008 (−0.05, 0.01) | −0.006 (−0.07, 0.02) | −0.002 (−0.04, 0.03) |

*Notes.* *** $p < 0.001$; ** $p < 0.01$; * $p < 0.05$. Path a = anthropomorphism to guilt. Path b = guilt to PEB (when anthropomorphism and the other emotions were also in the model). Path c = anthropomorphism to PEB (i.e., total effect). Path c′ = anthropomorphism to PEB (when guilt and the other emotions were also in the model; i.e., direct effect). Age, gender, and NEP were controlled for. ANS = Anthropomorphism of Nature Scale. IDAQ = Individual Differences in Anthropomorphism Questionnaire. NEP = New Ecological Paradigm Scale.

## 4. Study 3

I conducted Study 3 to address three additional issues.

The first issue concerns the measurement of emotions. It has been argued that single emotional words or labels fail to adequately capture emotional experiences [20]. Indeed, according to the factor analysis results, the items in Study 2 were not adequately differentiable from each other. Thus, in Study 3, I adopted a more comprehensive emotion measure, referring to not only emotional labels but also other emotional components (e.g., self-experiences, bodily sensations, action tendencies, motivational goals) [11,20].

Second, in Studies 1 and 2, pro-environmental behavior was measured with self-reported intention or past performance; no actual behavior was observed. This may present a threat to the validity of the findings because, for example, people may over-report their intention to perform pro-environmental behavior due to the concern of social desirability. Also, the similarity of the measurement method (i.e., self-report) across all key variables may artificially inflate their observed relationships [33]. Thus, in Study 3, I added a behavioral measure (in the form of an actual donation to an environmental organization) [50].

Third, recent studies have shown that relationships among environmental variables are not necessarily replicated across different societies and cultures [51–54]. In particular, some have argued that guilt, given its interpersonal nature, might have a stronger effect on behavior in collectivistic societies [55]. To establish the cross-cultural generalizability of the findings, I used a British sample instead of a Hong Kong Chinese sample in Study 3. Hong Kong and the United Kingdom are similar in terms of economic development (both ranking among the top 25 in the world in terms of GDP per capita) [56], but differ substantially from each other on individualism–collectivism [57].

### 4.1. Participants

The sample was recruited through an online panel called Prolific Academic (https://www.prolific.ac/). A recent study [58] showed that participants from Prolific Academic produced data quality that was higher than participants from other online panels, and were more naïve to research tasks and diverse in terms of backgrounds. In total, 255 individuals completed the study. They were compensated £1.30 for their participation. Following other researchers' advice regarding the use of online samples [59], three checks were adopted to ensure data quality. First, an attention check was included. Second, to ensure that participants were British, a question regarding the length of their residence in the United Kingdom was included. Third, to make sure participants were naïve to the study, participants were asked if they had participated in studies on similar topics previously. Eighteen participants were dropped from further analyses for failing either or all of these checks. The final sample comprised 233 participants (75 males and 158 females). Their age ranged from 18 to 70 years (mean = 25.64 and S.D. = 5.55 years), and had been living in the United Kingdom for 12 to 70 years (mean = 25.44 and S.D. = 5.71 years). Based on what I reported at the beginning of Study 1, this sample size was sufficient for achieving 0.80 power.

### 4.2. Measures

Participants completed the following measures. As in Studies 1 and 2, the ANS was used ($\alpha = 0.88$). The measure of environmental emotions was similar to that in Study 2. However, a more elaborate list of items that covered various emotional components was used. These items were written based on past emotion studies [11,20]. They referred to not only subjective feelings, but also to thoughts, bodily reactions, action tendencies, motivational goals, etc., associated with the emotions. To keep the measure within a reasonable length, shame was dropped. There were 11 items for guilt, 11 items for anger, and eight items for anxiety. Participants responded to these items on a 7-point scale (1 = strongly disagree to 7 = strongly agree). An exploratory factor analysis revealed the expected three-factor solution (Eigenvalues = 12.31, 2.86, and 1.87; explained variance = 41.03%, 9.52%, and 6.24%). After

dropping the items with double loadings, 10 items for guilt (e.g., "I want to be forgiven"; $\alpha = 0.92$), six items for anger (e.g., "I want to get back at someone"; $\alpha = 0.87$), and five items for anxiety (e.g., "I think that the world is not safe"; $\alpha = 0.83$) were retained.

Participants also completed three measures of pro-environmental behavior. The first two intention measures referred to private-sphere behavior (12 items; e.g., buy products with less packaging) and public-sphere behavior (12 items; e.g., boycotting companies that are not environmentally friendly), respectively. The items were adopted from a 25-country study on pro-environmental behavior [60]. Participants indicated how likely they were to engage in each action on a 5-point scale (1 = not at all to 5 = extremely likely). These two intention measures were reliable ($\alpha$s = 0.86 and 0.93, respectively). As for the third measure, I presented an opportunity for participants to donate money to an environmental organization. At the end of the study, I told participants that in addition to the promised compensation of £1.30, they had just earned a bonus of £0.50, and they could choose to either take all of the bonus, or donate part or all of it to an environmental organization (which was the World Wide Fund for Nature, based on [50]). Participants then indicated the amount out of the bonus they would like to donate using a slide bar. Participants were then paid the compensation plus the part of the bonus they did not donate.

A mean score was computed for each measure (except for the donation measure).

### 4.3. Results and Discussion

Table 5 presents the correlations among the variables. As hypothesized, anthropomorphism was significantly associated with guilt; it was also significantly associated with anger. Anthropomorphism was significantly associated with the two intention measures but not donation. Guilt was significantly associated with all three measures of pro-environmental behavior; the same was true for anger and anxiety (except for the association between anger and donation), though apparently to a weaker extent.

**Table 5.** Descriptive statistics and inter-correlations of the key variables in Study 3.

| | | Mean (S.D.) | 1 | 2 | 3 | 4 | 5 | 6 |
|---|---|---|---|---|---|---|---|---|
| 1. | ANS | 4.30 (2.77) | | | | | | |
| 2. | Guilt | 3.78 (1.24) | 0.14 * | | | | | |
| 3. | Anger | 2.56 (1.16) | 0.24 *** | 0.50 *** | | | | |
| 4. | Anxiety | 4.90 (1.27) | 0.08 | 0.67 *** | 0.39 *** | | | |
| 5. | Private-sphere behavior | 3.67 (0.76) | 0.23 *** | 0.36 *** | 0.15* | 0.33 *** | | |
| 6. | Public-sphere behavior | 2.33 (0.93) | 0.27 *** | 0.50 *** | 0.37 *** | 0.38 *** | 0.59 *** | |
| 7. | Donation | 0.20 (0.22) | 0.04 | 0.34 *** | 0.10 | 0.31 *** | 0.17 ** | 0.25 *** |

*Notes.* *** $p < 0.001$; ** $p < 0.01$; * $p < 0.05$. ANS = Anthropomorphism of Nature Scale.

I then tested the hypothesized pathway from anthropomorphism, through guilt, to pro-environmental behavior. Age and gender were controlled for. Table 6 shows the results. Anthropomorphism significantly predicted guilt (Path a) and also anger. It also significantly predicted pro-environmental behavior intention but not donation (Path c). When anthropomorphism, guilt and the other emotions were all included in the model (Path c' and Path b, respectively), guilt consistently predicted pro-environmental behavior intention and donation, whereas anthropomorphism predicted only intention but not donation. Recent discussions suggest that for various reasons (e.g., presence of suppression effects, asymmetries of statistical power), an indirect effect might be detected even when the total effect of the predictor is non-significant [44]. Following the recommendations from a recent review [61], I proceeded to test the hypothesized pathway for donation too, even though the total effect

of anthropomorphism on donation was non-significant. Using the bootstrap test (5000 resamples), I observed that the hypothesized indirect effect through guilt (a X b) was significant (i.e., the 95% CI did not include zero) for both intention measures as well as donation. There was no significant indirect effect through the other two emotions at all.

**Table 6.** Results of regression analyses and tests of indirect effects in Study 3.

| | | Private-Sphere Behavior | Public-Sphere Behavior | Donation |
|---|---|---|---|---|
| | | | *Outcome in the Model* | |
| *Unstandardized regression coefficients (standard errors)* | | | | |
| Path a | ANS → guilt | 0.06 * (0.03) | 0.06 * (0.03) | 0.06 * (0.03) |
| | ANS → anger | 0.10 *** (0.03) | 0.10 *** (0.03) | 0.10 *** (0.03) |
| | ANS → anxiety | 0.04 (0.03) | 0.04 (0.03) | 0.04 (0.03) |
| Path b | guilt → PEB | 0.17 ** (0.05) | 0.28 *** (0.06) | 0.05 ** (0.02) |
| | anger → PEB | −0.07 (0.05) | 0.08 (0.05) | −0.02 (0.01) |
| | anxiety → PEB | 0.11 * (0.05) | 0.09 (0.05) | 0.03 (0.01) |
| Path c | ANS → PEB (total effect) | 0.06 *** (0.02) | 0.09 *** (0.02) | 0.004 (0.005) |
| Path c′ | ANS → PEB (direct effect) | 0.06 ** (0.02) | 0.06 ** (0.02) | 0.001 (0.005) |
| *Size of indirect effects (bootstrap test 95% CI)* | | | | |
| a X b | ANS → guilt → PEB | 0.01 (0.001, 0.03) | 0.02 (0.001, 0.04) | 0.003 (0.0002, 0.01) |
| | ANS → anger → PEB | −0.01 (−0.02, 0.001) | 0.008 (−0.002, 0.03) | −0.002 (−0.01, 0.001) |
| | ANS → anxiety → PEB | 0.004 (−0.001, 0.02) | 0.003 (−0.001, 0.02) | 0.001 (−0.0003, 0.004) |

Notes. *** $p < 0.001$; ** $p < 0.01$; * $p < 0.05$. Path a = anthropomorphism to guilt. Path b = guilt to PEB (when anthropomorphism and the other emotions were also in the model). Path c = anthropomorphism to PEB (i.e., total effect). Path c′ = anthropomorphism to PEB (when guilt and the other emotions were also in the model; i.e., direct effect). Age and gender were controlled for. ANS = Anthropomorphism of Nature Scale. PEB = pro-environmental behavior.

In summary, with a more stringent measure of emotions, an additional measure of actual behavior, and a culturally different sample, the findings replicate those in Studies 1 and 2 and support my hypothesis.

## 5. General Discussion

### 5.1. Understanding Environmental Guilt

The present research offers an integrated solution to the two unresolved issues in the understanding of environmental guilt discussed earlier. First, some individuals experience guilt for the degradation of the environment, whereas some others do not. In *Greendex*, there was an almost equal split between respondents who reported the feeling, respondents who were opposed to it, and respondents who were neutral [1]. What is the reason for these individual differences? Second, if guilt is indeed a predominantly interpersonal phenomenon, as emotion research has documented [8], how is it possible for some individuals to experience guilt about the degradation of the non-human environment? The present findings suggest that anthropomorphism of nature allows individuals to attribute moral patiency to the environment; as a result, the cognitive template of harm applies and guilt becomes possible. Because there is inter-individual variation of anthropomorphism of nature, there are individual differences in environmental guilt. In other words, some individuals are more likely than others to view nature in an anthropomorphic manner; these individuals are more likely to consider the environment as a moral patient and consequently experience environmental guilt. They are also more motivated to take actions to protect the environment.

The present investigation also contributes to the study of environmental guilt by highlighting several methodological considerations. In the three studies, measures were taken to control for some potential third variables (e.g., age, pro-environmental worldview); standardize the environmental problems participants thought about; broaden the measure of emotions from using single-word labels only to including various emotional components [20]; measure pro-environmental behavior comprehensively by covering both private-sphere and public-sphere types [47] and including both

intention and actual behavior [42]; minimize the item context effects by using the prospective design [50]; establish cross-cultural robustness of the findings [52,55]; and control for other negative emotions that people might experience in the face of environmental degradation [14]. Together, these measures have helped establish the validity and generalizability of the findings. The conceptual replication of the findings across the three studies also echoes the call for attention to the issue of replicability in psychological studies [31]. It is recommended that future studies on environmental guilt also consider these methodological issues. In particular, comparing the results between Study 2 and Study 3, broadening the emotional measures seems beneficial. The various negative emotions became more differentiable from each other with the improved measures, and the findings regarding guilt became more robust. In future studies, researchers should consider using the improved measure of environmental guilt and refrain from measuring it with single-word labels.

The association between anthropomorphism and environmental guilt was consistent across the three studies. However, its size was generally small to medium (with *r* in the range of 0.20), implying that there are plausibly other predictors of the experience of environmental guilt. Notably, the role of personal responsibility has been identified in past studies. Guilt is unlikely to arise if an individual believes that he/she has nothing to do with the harm [11,45]. For example, in one study [7], it was observed that their participants reported a much stronger feeling of guilt when they had recalled an unpleasant event for which they were responsible (vs. one for which they were not responsible). In the environmental domain, people who believe that they have contributed to environmental problems experience more guilt [15,16]. The role of personal responsibility was not directly addressed in the present investigation. In the three studies, personal responsibility was presumed by acknowledging the contribution of humanity to environmental problems in the emotion measures [13]. Ideally, future studies should explicitly consider the effects of anthropomorphism and personal responsibility simultaneously. It seems logical to expect an interaction effect between these two factors. That is, anthropomorphism should have no effect on guilt among people who do not perceive any personal responsibility in environmental problems. Also, personal responsibility should have a lesser effect on guilt among individuals who do not anthropomorphize the environment.

My proposed account with regard to the role of anthropomorphism should not be seen as the only solution to the two unresolved issues aforementioned. As noted in the introduction, there are two alternative accounts. First, people may feel guilty for the harm caused to people affected by environmental problems (e.g., future generations). This account is still in line with the interpersonal essence of guilt. Second, guilt is not exclusively interpersonal. Apparently, these two alternative accounts cannot fully explain the findings reported (particularly the association between anthropomorphism of nature and environmental guilt). Nevertheless, I reckon that the proposed account based on the concept of anthropomorphism and these two alternative accounts do not discredit each other. In fact, it appears to be theoretically interesting to speculate that environmental guilt is actually a blend of several types of guilt: environment-oriented guilt (i.e., guilt for the harm done to the non-human environment), human-oriented guilt (i.e., guilt for the harm done to other people affected by environmental problems), and non-interpersonal guilt (e.g., guilt for not meeting one's own standards for living an environmentally friendly lifestyle). Future studies should consider how to differentiate between these contents of environmental guilt and explore to what extent they have different antecedents and different behavioral consequences.

## 5.2. Understanding the Human–Nature Relationship

People may refuse to attribute human qualities to some members of humanity [24]. Obviously, this psychological tendency, or dehumanization, violates the objective reality; nevertheless, it is psychologically real and impactful. For example, studies have shown that dehumanization is one of the mechanisms behind racial prejudice and discrimination [62]. Similarly, anthropomorphism can hardly be said to be an objective description of the non-human natural world; nonetheless, it is also psychologically real and impactful [9]. As the present investigation shows, with anthropomorphism,

environmental guilt and its behavioral consequences become possible. That people might consider nature as humanlike implies that there are perhaps some psychological processes that the study of the human–nature relationship has yet to fully understand [10]. With anthropomorphism, humans might relate to the environment in a way that is similar to how they relate to other humans. I propose that researchers should try to describe and understand these processes with reference to existing psychological concepts regarding interpersonal relationships. Recent studies have illustrated the viability of this proposed approach. For instance, a study [10] showed that people experience a sense of social connectedness to nature when nature is anthropomorphized. Also, it has been found that individuals who anthropomorphize nature exhibit empathy, an emotional reaction that is typically observed as a response to the suffering of other humans, when they consider environmental degradation [29,30]. The present investigation adds more evidence to this research approach by showing that guilt, a supposedly interpersonal emotion, can be used to describe some individuals' responses to environmental degradation. This approach represents an innovative way to integrate insights from social and interpersonal psychology with environmental psychology.

*5.3. Limitations*

The primary goal of this investigation is to understand the individual differences of environmental guilt. Such individual differences are very clearly illustrated in the *Greendex* study, and are implied in the anecdotal cases cited at the beginning, as well as empirical findings in psychological research discussed in the introduction. Because the phenomenon of interest concerns individual differences, the approach taken in the investigation is cross-sectional and correlational, rather than experimental. Accordingly, to what extent anthropomorphism has a causal effect on the experience of environmental guilt and hence pro-environmental behavior is still unknown. In his classic paper, The Two Disciplines of Scientific Psychology, Lee Cronbach [63] noted that the experimental method and the correlational method each has its own virtues; the correlational method is in general more multivariate, and the findings it generates can guide experimentation. Thus, an integration of both methods is ideal. Now, with the cross-sectional findings regarding individual differences reported here, the next studies should be aimed to look for experimental evidence of the effects of anthropomorphism of nature.

There is room for further methodological improvements even if the correlational method is adopted in the next studies. For example, in the three studies reported, to address the potential "third variable" problem, age, gender, and pro-environmental worldview were controlled for. However, it is obvious that there perhaps exist many other potential third variables, and the studies reported in this research can hardly control for all of them. Some constructs that capture the psychological connection between an individual and nature (e.g., connectedness to nature, anthropocentrism versus ecocentrism) [10,64,65] and some constructs that capture fundamental differences between individuals (e.g., values, personality) [66,67] are known to be associated with a general pro-environmental psychological orientation. Controlling for these constructs will provide further evidence to the validity of the association between anthropomorphism of nature and environmental guilt.

Another area for improvement concerns the representativeness of the samples. In the present investigation, efforts were exerted to diversify the samples used. Both students and working adults were used, and participants from different cultural backgrounds were included. Still, the samples were convenience-based; they were hardly representative of the population. To what extent the present observations can be generalized to the population therefore remains to be addressed. It is suggested that in future studies a nationally representative sample is recruited. Alternatively, it will be useful to incorporate items regarding anthropomorphism and guilt into such international survey studies as *Greendex* (in which representative samples are typically used).

*5.4. Practical Implications*

Given the present findings, one may feel tempted to recommend the use of anthropomorphic appeals [68], as well as guilt appeals [13], in environmental messages. Past studies have demonstrated

that it is often challenging to moralize environmental issues and motivate public engagement in the mitigation of environmental problems [69]. The present research provides a framework that can address this challenge. If people do not perceive any moral patients in environmental degradation, it is unlikely that the cognitive template of harm will apply, and that people will experience moral involvement and take moral actions. Thus, it seems reasonable to suggest that environmental communicators need to clearly and explicitly identify the moral patients (e.g., anthropomorphized nature) in environmental degradation in their messages.

Having said that, I emphasize that the use of anthropomorphic appeals and guilt appeals must undergo further scientific scrutiny. As discussed earlier, there are concerns that promoting anthropomorphism may obfuscate people's accurate understanding of the natural world [35,36] and generate undesired effects with regard to conservation [37,38]. Furthermore, some have questioned whether the use of negative emotions such as guilt is exerting an unjustifiable moral burden on the audience [6] and whether such emotional appeals will backfire [69]. It is also noteworthy that guilt does not necessarily motivate reparative behaviors. Research has shown that people may manage their guilt in many other ways, such as scapegoating [70], religious confession [71], and moral disengagement strategies (e.g., diffusion and displacement of responsibility, misrepresenting or distorting the consequences of harmful action, or reframing harmful action as serving some worthy causes) [72]. I believe that more evidence on how these different concerns can be addressed is needed before recommendations for the use of anthropomorphic appeals and guilt appeals are made.

## 6. Conclusions

To conclude, the present investigation aimed to understand environmental guilt, a common experience among the public, by identifying the role of anthropomorphism. It was observed that individuals who anthropomorphize nature are more likely than those who do not to experience guilt when considering environmental degradation, and this guilt feeling, in turn, is associated with more engagement in pro-environmental behavior. The pattern was mostly consistent across the three studies with different samples, methodological procedures, and measures. Theoretically, this observation improves the existing understanding of environmental guilt by suggesting a factor that underlies the individual differences of the feeling and demonstrating how it is possible for some individuals to experience guilt, a predominantly interpersonal emotion, for the degradation of the non-human environment. The present findings also add evidence to the theoretical possibility of describing and understanding the human–nature relationship with reference to existing psychological knowledge regarding interpersonal relationships. Practically, these findings potentially support the use of anthropomorphic appeals and guilt appeals in environmental messages, although this notion has yet to be more thoroughly tested in future studies.

**Supplementary Materials:** The following are available online at http://www.mdpi.com/2071-1050/11/19/5430/s1: Study 1 Measures, Study 1 Data, Study 2 Measures, Study 2 Data, Study 3 Measures, and Study 3 Data.

**Funding:** The work described in this study was partially supported by a grant from the Research Grants Council of the Hong Kong Special Administrative Region, China (Project No. HKUST645311) and a grant from the Hong Kong University of Science and Technology, Hong Kong (Project No. SBI15HS07).

**Conflicts of Interest:** The author declares no conflict of interest.

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
