# Peer review of "Anthropomorphism of Nature, Environmental Guilt, and Pro-Environmental Behavior"

_sustainability, doi:10.3390/su11195430_

Round 1

Reviewer 1 Report

Kim-Pong Tam ‘Anthropomorphism of Nature, Environmental Guilt, and Pro-Environmental Behavior’ review for Sustainability.

Response/Evaluation:
This is a very well written and engaging piece of scholarship which unfortunately needs to be re-worked to meet academic standards of coherence. The focus of study of environmental guilt has been an ongoing preoccupation within environmental studies and this paper certainly provokes some useful engagement. The author asserts in the abstract that ‘It appears that individuals who view nature in anthropomorphic terms are more likely to feel guilty for environmental degradation, and they take more environmental action as a result’. What a big claim to make? But how to prove this, especially if you don’t have a ‘control group’. Also by not teasing out alternatives to anthropomorphism you are not fully exploring the concept; maybe for instance unpacking feelings of biocentrism would be a good place to start. Furthermore, there are lots of environmental critiques focused on the over-anthropomorphizing nature – as in ‘the Bambi effect’ in film animation, which was designed to encourage humans and especially children to empathise with animals and feel guilt/shame at their environmental destruction? Some environmentalists however suggest this supports an over-sentimentalising of nature.  For example see some of these contradictory debates in Hollywood Utopia: Ecology in Contemporary American Cinema (Brereton 2005).

Furthermore, I would take issue with the theorising of the concept of ‘guilt’, which is not always outward directed, but for many religions for example is frequently self and inward-directed. While methodologically the author(s) ‘tested whether orthomorphism of nature was associated with environmental guilt, and whether guilt in turn was associated with self-report participation in the event in the past and intention to participate in it in future.’ I would question how can you possibly collapse all these concepts together to ‘prove’ a pregiven thesis? The author tries valiantly to address some of these weaknesses by ‘re-doing’ the experiment two further times – but with different cohorts of participants.

I particularly like the assertion made later in the paper: ‘In fact, it is theoretically interesting to speculate that environmental guilt is actually a blend of several types of guilt: environmental oriented guilt – human oriented guilt – and non-interpersonal guilt – e.g. guilt for not meeting one’s own standards of living an environmentally friendly lifestyle.’ I would almost start the paper with this nuanced assertion as a possible working hypothesis, rather than foregrounding a simple [even naive] concept of guilt which can be unpacked more easily.

Overall, in spite of these caveats, this is a solid enough working paper, but probably is trying to do far too much and remains under-cooked in places. In particular there are lots of definitional gaps I would like to be teased out, beginning with the anthropomorphic literature and its opposites/alternatives as well as lots of methodological gaps – basically how to stop trying to do too much at times and in particular clarifying how the 3 studies work as ‘one’ thesis. Overall, I still believe this could develop from a very engaging work in progress. All of these observations are made in good faith to help improve the robustness of the paper. It remains for the author to respond to and address in a re-write for a re-submission. Unfortunately, in my judgement at present the MS remains a work-in-progress.

Abstract –

Feeling guilty for the happening [this is a wrong word in this context? – maybe you could use ‘occurrence’] of environmental problems is not uncommon, but not everyone experiences it. Why is there such individual difference? [across the world? Or only evident within this paper’s sample study and geographical orbit?]… how is it possible for some individuals to feel guilty for the degradation of the non-human environment? [and others not, one presumes?]…. It appears that individuals who view nature in anthropomorphic terms are more likely to feel guilty for environmental degradation, and they take more environmental action as a result. [What a big claim to make? But how to prove! If you don’t have a ‘control group’ – and/or teasing out the alternative to anthropomorphism – maybe for instance unpacking counter/feelings of biocentrism? All in all, this remains the biggest criticism and question for the author to respond to; if not addressing in a re-write for this submission?

Introduction

Good evidence of extensive reading around academic studies of feeling guilty around the environment. … and ‘therapy of environmental guilt’. Annina Rust as artist-innovator, whose work reminds me of Christian Martyrs or other extreme anti-materialist religious cults?

… guilt is essentially an interpersonal phenomenon [Not strictly so I would suggest for extreme religious puritanical Early Christians for example – who as believers had strict codes of sin/shame, often not emanating from interpersonal but from private thoughts or actions etc.

Environmental Guilt

The feeling of guilt typically motivates apologies, attempts to reduce the harm, and actions to make up for one’s misdeeds and compensate for the victim’s loss…

Studies have shown that environmental guilt is indeed associated with pro-environmental behaviour….[Yes, this is a common thread within much of the environmental literature.]

Another study [2012] showed that Americans who reported a stronger feeling of guilt for their country’s contribution to environmental problems and lack of actions to protect the environment reported stronger willingness to engage in eco-friendly behaviour. [How this has ‘changed’ with the rise of climate sceptic Trump and his reneging on Paris agreement etc?].

Guilt typically involves concern over exclusion by other people and very often motivates behaviour that can repair or enhance social relationships (e.g. apology, reparations, compensations). [Not always I would counter – some guilt is actually self-directed and self-caused?]

… If guilt is essentially an interpersonal emotion, how is it possible that some individuals feel guilty for the degradation of the non-human environment? Also, what explains the individual difference of this feeling? P.2

p.3 linking human foetuses and animals, [by all accounts is taking on hugely contentious issues, which are not always seen on the same scale – depending on where culture/ethics is framed, regarding the central importance of human’s vis a vis other sentient beings etc.]

Note: lots of environmental critiques of over-anthropomorphizing nature – as in ‘Bambi effect’ in film animation, which was designed to encourage humans and especially children to empathise with animals and feel guilt/shame at their environmental destruction?

Some environmentalists however suggest this supports an over-sentimentalising of nature.  For example see some of these debates in Hollywood Utopia: Ecology in Contemporary American Cinema (Brereton 2005).

Case study 1 – of Earth hour – symbolic act of turning lights off for an hour – using sample of working adults in Hong Kong…

I tested whether orthomorphism of nature was associated with environmental guilt, and whether guilt in turn was associated with self-report participation in the event in the past and intention to participate in it in future. [How can you possibly collapse all these concepts together to ‘prove’ a pregiven thesis?]

Using 176 adult [one would hope so?] staff members from a university in Hong Kong…

Somewhat of a bold statement to say that the association between anthropomorphism of nature and environmental guilt has never been studied? - has at least tangentially been explained in lots of environmental film studies for instance] p.3

Measures of analysis seem very advanced – using an 11-point scale – seems excessive…

How to handle danger of suggesting answers regarding how you frame questions: ‘I am regretful about humans’ lack of performance on environmental protection’ and ‘I feel guilty for how little humans have done to improve the quality of the environment’. [Best, usually not to ‘spell out’ what your ‘preferred’ responses might be and find other ways to ‘illustrate’ or alluded to, certainly would be better. Basically not using ‘leading questions’.

As hypothesized, anthropomorphism was positively associated with guilt. [how did you deal with any staff who did not embrace/recognise this conjunction? Surely some (critical) academics did not concur with your thesis?

Study 2 – [how soon after the last experiment was this carried out – using improvements to last experiment.. adding anger, anxiety and shame – which is certainly an improvement…

BUT not using academics as focus - this time using students – so can’t compare/contrast with study 1?

And the students participated for course credit! Readers would like to know some details on type of students were involved and what they were studying?

Far too crude to suggest on p.7 that Study 1 was successfully replicated?

Nevertheless, good evidence of statistical analysis of this project.

Study 3 – addressing measurement of emotions – this is another huge study in itself – helping to tease out if not define emotional literacy?

Using possibility of a small donation to measure environmental behaviour…

Sample of 255 individuals who were give 1.30 sterling for their participation ..

So many variables not sure one can rely on any of them..

But p.15 however, solid analysis and good knowledge of research in the field..

I like the assertion: In fact, it is theoretically interesting to speculate that environmental guilt is actually a blend of several types of guilt: environmental oriented guilt – human oriented guilt – and non-interpersonal guilt – e.g. guilt for not meeting one’s own standards of living an environmentally friendly lifestyle. I would almost start with this nuanced assertion as a possible working hypothesis, rather than foregrounding a simple [even naive] concept of guilt … p.15

With anthropomorphism, humans might relate to the environment in a way that is similar to how they relate to other humans. I propose that researchers should try to describe and understand these processes with reference to existing psychological concepts regarding interpersonal relationships… study shows that people experience a sense of social connectedness to nature when nature is anthropomorphized. [p.16 – Elemental literary and filmic eco-analysis – see ISLE journal and so many studies of therapeutic nature of the environment to affect [anthropomorphic] agenda of human audience engagement?

Overall this is a solid enough paper, but probably trying to do far too much and is under-cooked in places. In particular there are lots of definitional gaps I would like to tease out, beginning with the anthropomorphic literature and its opposites/alternatives as well as lots of methodological gaps – trying to do too much at times and clarifying how the 3 studies work as ‘one’ thesis… Overall, I feel this is still a very engaging work in progress. 

Reviewer 2 Report

Dear author,

Thank you very much for your manuscript. I find the research question interesting and important, the theoretical frame and methodological approach appropriate and the analysis clear.

The only comment I would like to make concerns the concept of anthropomorphism, where I would like to invite you to say just a little bit more about it.

The background to my comment is the following. I deal with anthropomorphism in the field of human-animal studies, where it is understood as a coin with two sides. It can be a naiv, unreflected and inappropriate projection of human characteristics onto animals; but it is also the basis of understanding animals: feeling what other beings might undergo in a specific situation. So for me your definition of anthropocentrism (and the measurement you use) seems too negativ and onedimensional. I do not know, how this might be transferred to the environment or nature - there are important differences which you also demonstrate in your factor analysis that distinguishes between anthropomorphism for nature and for animals. However, there might be an aspect in there that can be formulated in a more positive way than you put it. My thinking here expresses more curiosity and wonder than a critique.

But it might help readers anyway to add two or three sentences on your understanding and the more general discourse of anthropomorphism in the context of the environment.

Wishing you all the best!

Round 2

Reviewer 1 Report

See File attached... 

Really works well I think - addressing many of issues raised... 
